# Proteomics Readjustment of the *Yarrowia lipolytica* Yeast in Response to Increased Temperature and Alkaline Stress

**DOI:** 10.3390/microorganisms9122619

**Published:** 2021-12-18

**Authors:** Varvara Y. Sekova, Leonid I. Kovalyov, Marina A. Kovalyova, Natalya N. Gessler, Maria A. Danilova, Elena P. Isakova, Yulia I. Deryabina

**Affiliations:** A.N. Bach Institute of Biochemistry, Research Center of Biotechnology of the Russian Academy of Sciences, Leninsky Ave. 33/2, 119071 Moscow, Russia; beauveria606@gmail.com (V.Y.S.); kovalyov@inbi.ras.ru (L.I.K.); M1968@mail.ru (M.A.K.); gessler51@mail.ru (N.N.G.); ms.maria.danilova@gmail.com (M.A.D.); yul_der@mail.ru (Y.I.D.)

**Keywords:** yeast, heat shock, extreme ambient pH, combined stress, proteomic readjustments

## Abstract

Yeasts cope with a wide range of environmental challenges using different adaptive mechanisms. They can prosper at extreme ambient pH and high temperatures; however, their adaptation mechanisms have not been entirely investigated. Previously, we showed the pivotal role and flexibility of the sugar and lipid composition of *Yarrowia lipolytica W 29* upon adaptation to unfavorable conditions. In this study, we showed that extreme pH provoked significant changes in the cell wall proteins expression, with an increase in both the chaperones of heat shock protein HSP60 and some other proteins with chaperone functions. The mitochondria activity changes inducing the VDAC and malate dehydrogenase played an essential role in the adaptation, as did the altered carbohydrate metabolism, promoting its shift towards the pyruvate formation rather than gluconeogenesis. The elevated temperature led to changes in the cell wall proteins and chaperones, the induced expression of the proteins involved in the cell structural organization, ribosomal proteins, and the enzymes of formaldehyde degradation. Moreover, the readjustment of the protein composition and amount under combined stress indicated the promotion of catabolic processes related to scavenging the damaged proteins and lipids. Under all of the stress conditions studied, the process of folding, stress resistance, redox adaptation, and oxidative phosphorylation were the dominant pathways. The combined chronic alkaline and heat stress (pH 9.0, 38 °C) led to cross-adaptation, which caused “switching” over the traditional metabolism to the adaptation to the most damaging stress factor, namely the increased temperature.

## 1. Introduction

The ability of a living organism to prosper under different kinds of stress is associated with the changes in the gene expression profile followed by significant readjustment at the molecular and biochemical levels. Some abiotic environmental factors, namely extreme ambient pH, high temperatures, and osmolarity, lead to the most remarkable influence on the growth and development of yeast cells. The adaptation to these conditions mainly includes the increased synthesis of some stress proteins and the essential rearrangement of cell metabolism. In addition, it leads to compositional changes in membrane lipids and the accumulation of various osmolytes and protective agents in the cytosol [1]. The extremophile *Yarrowia lipolytica* is a unique yeast which thrives in a wide range of environmental challenges such as ambient pH from 2.5 up to 9.5 [2,3], high salinity, dry or hydrophobic substrates [Liu et al., 2015], and elevated temperatures up to 38 °C [4]. 

A yeast’s adaptation to various stresses includes some metabolic pathways, namely changes in fatty acid synthesis, their unsaturation degree, modification of membrane phospholipid components, some alterations in the cAMP levels, the mitochondrial activity, and ATP synthesis [5]. The exclusive mechanism of a general response to the stress in a yeast is supposed to protect a cell against harmful environmental challenges [6]. 

Nevertheless, metabolic readjustment due to the changes in the cell proteomic composition is a keystone of the adaptation to any stress. In the paper by [7], the authors showed that the proteome of *Y. lipolytica* yeast WSH-Z06 significantly changed under acid stress during the production of *α*-ketoglutaric acid. The proteins maintaining intracellular pH homeostasis, which greatly increased at acidic pH, were divided into several groups according to their functions in the cells. The authors speculated that the ATP synthesis activates due to membrane hyper-polarization upon cytoplasm acidulation. It causes an increase in ROS generation, and the cells need antioxidant components and chaperones to protect the cell against ROS. To induce metabolite exchange between the cytosol and mitochondria, the VDAC level in the cells increases. The *Y. lipolytica* proteome under alkaline stress compared to that under normal conditions was studied before [8]. The authors showed that the level of some mitochondrial proteins, namely *α*-ketoglutarate dehydrogenase, malate dehydrogenase, some outer membrane proteins (porin VDAC), and the respiratory chain components, notably increased in the cells under alkaline conditions (pH 8.5). Moreover, the level of the cytoprotective proteins, in particular, Cu/Zn-superoxide dismutase (SOD), rose. However, the authors failed to detect an increase in the Na^+^-dependent ATPase level in the cells, which would presumably provide the phosphate transport upon the cytoplasmic membrane depolarization. This was shown in the paper by [9], where the response of the *Y. lipolytica CICC 1675* proteome to osmotic stress at a high NaCl concentration (up to 70 g/L) was under investigation. Among the 55 up-regulated proteins during osmotic stress, TCA enzymes, respiratory chain components, antioxidant enzymes, heat shock proteins, and carbohydrate metabolism enzymes responsible for the osmolytes synthesis were revealed.

Several studies indicating the general response to stress in the *Y. lipolytica* yeast have previously been published [10]. They showed cross-adaptation between some stressors such as heat shock, osmotic stress, ethanol, and induced oxidative stress. The authors found the ROS scavenging enzymes such as SOD, catalase, alternative mitochondrial oxidase, and glutathione reductase to be the key components of the general response.

In our recent studies, we researched the adaptation of the *Y. lipolytica* yeast to pH and heat stresses at the level of the glycom and lipidome, as well as the redox status of the cells [4]. The increased growth temperature was recorded to affect the carbohydrate and lipid metabolism more than the alkaline conditions did. Thus, mannitol was the dominating soluble carbohydrate of the cytosol upon the cultivation at 29 °C. However, at the increased temperature of 38 °C, mannitol was substituted by trehalose, possessing a thermo-protective effect [4].

In the present study, we show the adaptive proteomic readjustment of the *Y. lipolytica* yeast under thermal, alkaline, and combined stresses to identify the metabolic pathways involved in the adaptation of the cells.

## 2. Methods and Materials

### 2.1. Yeast Strain and Growth Conditions

Wild-type *Yarrowia lipolytica W 29* from CIRM Levures collection (France) was used. The culture was raised in batches of 100 mL in glycerol (1%)-containing medium of the following composition (g/L): Mg_2_SO_4_—0.5, (NH_4_)_2_SO_4_—0.3, KH_2_PO_4_—2.0, K_2_HPO_4_—0.5, NaCl—0.1, CaCl_2_—0.05). Then, 2M KP_i_ stock buffer was prepared by dissolving KH_2_PO_4_ anhydrous (272 g/L, Amresco Cat No. 0781), pH adjusted with 2M K_2_HPO_4_ to 6.0. Furthermore, 2M KP_i_ stock buffer was prepared by dissolving K_2_HPO_4_ anhydrous (342 g/L, Amresco Cat No. 0705), and the pH was adjusted with 2M KH_2_PO_4_ to 9.0. Both KP_i_ buffers were sterilized by autoclaving and added to sterilize the culture medium (ratio 1:40) just before inoculation. The yeast was cultivated at two ambient pH values of 5.5 and 9.0 and two temperatures of 29 and 38 °C on a rotary shaker at 150 r.p.m, as described by [4]. Absorbance (A) was assessed in cell suspension at the wavelength of 590 nm (A_590_) using a Specol-11 spectrophotometer (Carl Zeiss, Oberkochen, Germany). The yeast was raised in the stationary growth phase.

### 2.2. Sampling 

The biomass was harvested by centrifugation at 4000× *g* for 10 min. The cells were washed twice with ice-cold deionized water and eventually pelleted. To prepare protein extracts, 100 mg of the cell pellet was transferred to a vial containing 2 mL of lysis buffer (9 M urea, 5% β-mercaptoethanol, 2% Triton X-100, and 2% ampholytes, pH 3.5–10 (Sigma, Sigma-Aldrich, St. Louis, MO, USA) and thoroughly suspended. The sample was either immediately heated in a boiling bath for 3–5 min or placed on ice and sonicated in an ultrasonic desintegrator (MSE-Pharmacia, Stockholm, Sweeden) for 2 min (4 cycles, 30 s each). In both cases, the homogenate was clarified by centrifugation in a microfuge for 20 min at maximum speed. The pellet was discarded, and 100 μL of the clear supernatant was used for isoelectrofocusing (IEF).

### 2.3. Proteomic Analysis

The preparation of protein extracts, their fractionation by O’Farrell two-dimensional electrophoresis (2DE) with own modifications, and analysis of electrophoregrams were performed as described before [11,12]. At least three 2DE electrophoregrams were tested for each sample. The kits of highly purified recombinant proteins SM0661 (10–200 kDa), and SM0671 (10–170 kDa) (Fermentas, Waltham, MA, USA) were used as molecular-weight standards to calculate sample molecular weights. To visualize proteins, the polyacrylamide gel slabs were stained with Coomassie Blue R22 250 and then with silver nitrate according to well-described methods [13] and modified by the addition of 0.8% acetic acid to sodium thiosulfate. The stained gels were documented by scanning on an Epson Expression 1680 scanner (Epson, Suwa, Nagano, Japan), and densitometry was carried out using the Melanie software (GeneBio, Genève, Switzerland) according to the producer’s protocol. Theoretical values of M and pI were calculated using the “pI/Mw” software in open access at the ExPASy Proteomics Server (http://cn.expasy.org; accessed date: 10 October 2021) and data on amino acid sequences of corresponding proteins in SwissProt database considering the evidence for post-translational removal of signal sequences.

### 2.4. Protein Identification by Mass Spectrometry

The fractions chosen for identification were cut out from the 2DE gels and the proteins were hydrolyzed by trypsin. The extracted tryptic peptides were assayed using MALDI-TOF MS as described previously with some modifications [14]. The sample (0.5 μL) was mixed with the same volume of 20% acetonitril solution containing 0.1% of trifluoroacetic acid and 20 mg/mL of 2, 5-dihydroxybenzoic acid (Sigma), and was air-dried. Mass spectra were acquired on Ultraflextreme MALDI-TOF mass spectrometers (Bruker, Bremen, Germany) with a UV laser (336 nm) in the 500–8000 Da range in the positive ion mode calibrated using reference trypsin autolysis peaks. During MS/MS analysis, the mass spectra of fragments were acquired on a Bruker Ultraflex MALDI-TOF mass spectrometer (Bruker, Bremen, Germany) in the positive ion mode. The results of protein identification were processed with Mascot software (FlexControl 3.3, FlexAnalisis 3.3 и Biotools 3.2), Peptide Fingerprint option (Matrix Science, Boston, MA, USA), with an accuracy of mass measurement MH + of 0.01% (with a possibility to modify cysteines by acrylamide and methionine oxidation).

### 2.5. Statistical and Bioinformatics Analysis

The data presented are the average results of at least 3–5 replicates with a standard error of less than 5%. Analysis of variance was performed using Primer of Biostatistics v. 4.03 software. Significant differences between calculated values were evaluated using a paired *t*-test at the 5% level of probability. 

The molecular weights of protein fractions of two-dimensional electrophoresis were assayed according to their electrophoretic mobility in the second direction comparing them to the standard proteins-markers [10,15,16]. The results were confirmed and, in some cases, corrected according to the calibration curve made using a set of highly purified recombinant proteins with molecular weights between 10 and 200 kDa (Fermentas, Waltham, MA, USA) [17]. 

For the analysis of protein–protein interactions, differentially expressed proteins were searched against the STRING database (Search Tool for the Retrieval of Interacting Genes/Protein, http://string-db.org/; accessed on 26 October 2021) with the highest confidence score (score > 0.9). Only the interaction network with at least three proteins was shown, and the unconnected proteins or clusters with two proteins were not presented.

The measurements were carried out on the equipment of the Shared-Access Equipment Centre “Industrial Biotechnology” of Federal Research Center “Fundamentals of Biotechnology” Russian Academy of Sciences.

## 3. Results and Discussion

In this study, we compared the protein expression profile of the *Y. lipolytica* yeast cultivated at two ambient pH values (normal 5.5 and alkaline 9.0) and two growth temperatures (optimal, 29 °C and elevated, 38 °C) in various combinations. Previously, we showed that growth conditions of pH 5.5 and a temperature of 29 °C could be regarded as the normal ones, while the shift in ambient pH to the alkaline side (9.0) or the increase in the growth temperature up to 38 °C are the stress conditions. Growth at both alkaline pH and increased temperature is considered as being under combined stress [4]. The cultures grown under different conditions were raised at the stationary growth phase, extracted as described in the Materials and Methods, and the extracts were separated by O’Farrell two-dimensional protein electrophoresis. Then, the protein spots, which under different conditions significantly varied in their relative amount, were identified using MALDI-TOF mass spectrometry for the determination of differentially expressed proteins (DEPs). Sixty-six protein spots were successfully identified using mass spectrometry from the cells grown under different conditions. Comparative proteomics was used to evaluate the response of the yeast to the stress conditions. Figure 1 shows the results of their comparison, and the citation to the database is presented in Appendix A. Appendix A presents the full protein list.

Six unique proteins were revealed in normal conditions (Figure 1 and Appendix A). Among them, there is one protein of oxidative phosphorylation, the α-subunit of ATP synthase (YALI0F03179p); two structural proteins, mannoprotein of the cell wall Pir1 (YALI0B20306p) and profilin (YALI0B07183p), as well as thiol-specific peroxidase (YALI0A19426p) required to maintain cell redox homeostasis (Table 1). This set of proteins is associated with active cell growth and division (Appendix A). In total, we identified 51 expression proteins showing the most altered abundance in the normal and different stress conditions. Figure 1 demonstrates the experimental results of 2-DE gels.

### 3.1. Changes in the Yeast Proteomics at Alkaline Ambient pH

Table 2 shows the results of mass spectrometric identification of the proteins from the cells grown under alkaline stress. Twelve original proteins were expressed in these conditions (Table 2, Appendix A): two proteins associated with the ubiquitination, namely the SCF subunit of the E3 ubiquitin ligase complex (zone 29, YALI0A10879p) and an endopeptidase activator (zone 24, YALI0B09339) (Appendix A). Some other proteins expressed under alkaline pH are the members of the HSP family, namely small 20-kDa HSP (zone 35, YALI0C03443p). Besides that, we could identify the β-subunit of the immature polypeptide-associated complex (zone 36, YALI0F08393p). NADPH dehydrogenase (zone 27, YALI0B07007p) involved in purine metabolism, bis (5′-adenosyl) triphosphatase (zone 30, YALI0E32736p), (zone 27, YALI0E19723p), and calcium binding protein 31 (YALI0E03388p) are also expressed in significant amounts in these conditions (Figure 1 and Appendix A, Table 2). Moreover, the uncharacterized protein of YALI0D00451p (zone 25) and a heat shock protein 70-related protein SSC1 mitochondria precursor (zone 41, YALI0C17347p) were among the proteins induced by alkaline stress. However, four proteins in the proteomic profile of the *Y. lipolytica* yeast grown under normal conditions disappeared, namely zone 5 (mannoprotein (Pir1, YALI0B20306p) and zone 19 (peroxiredoxin protein family, YALI0A19426p) (Figure 1).

Comparative analysis of 2-DE gels revealed nine proteins with increased expression under alkaline stress (Figure 2 and Appendix A, Table 3). Among the up-regulated proteins at an alkaline pH, there are two folding proteins, namely 39 (YALI0C10230p, peptidyl-prolyl cis-trans isomerase) and zone 26 (YALI0F02805p, HSP60 heat shock protein—chaperone); two proteins of glycolysis: zone 22 (YALI0C06369p, glyceraldehyde-3-phosphate dehydrogenase) and zone 40 (YALI0F05214p, triose phosphate isomerase) (Appendix A). In addition, some mitochondrial proteins were identified, namely the outer membrane porin (zone 20, YALI0F17314p) and malate dehydrogenase (zone 21, YALI0D16753p); the ribosomal protein (zone 33, A0A1H6Q0M6), and the protein involved in the nucleotides synthesis (zone 35, YALI0F09229p) (Appendix A). Among the proteins down-regulated under alkaline stress, we obtained three proteins of oxidative phosphorylation, namely 29, YALI0D2202p; 30, YALI0E19723p, and 31, YALI0E10144p (Appendix A). The protein of cell wall synthesis (zone 23, YALI0B03564p), 60-S ribosomal ubiquitin (zone 37), and thioredoxin YALI0F01496p (zone 32) also significantly decreased in their amounts (Table 4, Appendix A). 

Eight protein spots showing the abundance at least 1.7-fold or more in the alkaline conditions were identified (Figure 3), namely zones 20 (mitochondrial porin VDAC), 21 (malate dehydrogenase), 22 (glyceraldehyde-3-phosphatedehydrogenase), 39 (peptidyl-prolyl cis-trans isomerase, modified peptidyl-prolyl cis-transisomerase), and 40 (triose phosphate isomerase) (Figure 3). The expression of proteins 40, 22, and 26 showed the most increase, namely by 3.3, 3.5, and 3.9 times, respectively (Figure 3). The proteins of 23 (1, 3-beta-glucanosyltransferase) and 37 (ubiquitin) decreased in the spot volumes in the electrophoretogram by 0.4 and 0.68 times, respectively (Figure 3). Cu/Zn SOD YALI0E12133p was the only protein with an unchanged level of expression (zone 16, Figure 1, Appendix A).

The structural protein of YALI0B20306p (zone 5, cell wall mannoprotein) is one of typical proteins for the normal conditions (Table 1). The yeast cell wall is known as the main barrier providing cell osmotic stability and protects it from an alkaline environment. Its composition mainly consists of neutral carbohydrates (70%), amino sugars (7%), proteins (15%), lipids (5%), and phosphates (0.8%). The *Y. lipolytica* cell wall usually contains mannan, *β*-glucan, and chitin as the main structural and storage polysaccharides. Unlike glucans and chitin, mannan undergoes alkaline hydrolysis [18,19,20]. Mannans are usually included in the wall outer part, while the inner surface is composed of chitins and glucans. Mannoproteins are covalently bound to cell wall mannans via either N-glycosidic or O-glycosidic bonds. The latter is more labile at an alkaline pH [21]. The expression of mannoprotein (Pir1) YALI0B20306p only in the yeast grown at pH 5.5 and 29 °C indicates its instability at an alkaline pH and 37 °C. Chaperones play a crucial role in the native conformation maintenance of proteins at osmotic stress. At an alkaline pH, we observed a 9.6-fold increase in the HSP60 heat shock protein YALI0F02805p (zone 26, Table 3, Figure 3) belonging to the HSP60 chaperone family, which support the synthesis and conformation of mitochondrial proteins. Previously, it was recorded that HSP60 chaperones are involved in folding and post-translational alterations of the mitochondrial proteins and imported proteins synthesized on the inner mitochondrial membrane [22,23]. The STRING database data (http://string-db.org/; accessed on 26 October 2021) show the interaction of YALI0F02805p with both HSP70 chaperones and the GroES chaperonin family and with ATP synthase subunit beta in the *Y. lipolytica* yeast (Figure 4A). The adaptation to alkaline conditions is likely to raise the participation role of HSP60 chaperones to support the protein conformation. At pH 9.0, two expression proteins, namely heat shock protein 70-related protein SSC1 mitochondrial precursor YALI0C17347p (zone 41, Table 2), interacting with HSP60 heat shock protein, and YALI0C03443p (small heat shock protein (HSP20) family), were identified.

However, the PPIase or rotamase YALI0C10230g (No 39, Table 3) increased, facilitating the cytoplasmic proteins’ folding. According to the STRING database (http://string-db.org/; accessed on 26 October 2021), the YALI0C10230g protein tightly interacts with the histone deacetylation system (YALI0B20262p and Histone deacetylase proteins) (Figure 4B), which regulates the transcription processes under these conditions [24]. The decreased share of ubiquitin-60S ribosomal protein L40 fusion protein (No 37, Table 4) regulating the ribosomes’ assembly and the expression of the 26S ribosomal regulatory unit YALI0B09339p (zone 24, Table 3) also indicate some disturbances in the ribosomes’ structural organization (some damage to the ribosome assembly during growth at an alkaline pH). One of the up-regulated proteins that was activated under alkaline conditions was YALI0C06369p, glyceraldehyde-3-phosphate dehydrogenase (No. 22, Table 3). The changes in carbohydrate metabolism play a key role in the adaptation of the *Y. lipolytica* yeast upon growth using glycerol at alkaline pH values. Previously, we showed a slight decrease in the storage lipids’ accumulation by 25–27% and cytosol soluble carbohydrates by 1.6 times under these conditions [4]. 

However, mannitol remained the main carbohydrate, while glucose made up no more than 5.5% of the soluble carbohydrate pool. According to [25], growth of *Y. lipolytica* using glycerol leads to a higher oxygen consumption compared to that using glucose. This suggests that glycerol should be involved in general carbon metabolism with the participation of glycerol-3-dehydrogenase related to the mitochondrial membrane. This speculation is confirmed by the fact that the protein tightly interacts with the glycolysis enzymes of phosphoglycerate kinase; YALI0D12400p, glucose-6-phosphate isomerase; YALI0F07711p, and triosephosphate isomerase; YALI0F01584p (Figure 4C). Glyceraldehyde-3-phosphate dehydrogenase interacts with triosephosphate isomerase, YALI0F05214p (No 40). Its expression increased 3.3-fold under alkaline conditions (Figure 3). We showed this is in a previous study [12]. 

The TPI activation and a shift of the dihydroxyacetone phosphate conversion into glyceraldehyde-3-phosphate followed by its transformation into glycerate-1,3-bisphosphate and NADH formation at the background of GAPDH activation were revealed. So, we could expect a slight decline in metabolites’ participation in gluconeogenesis, but maintaining a general scheme focus to form carbohydrates (mannitol) and lipids, as shown before. Triose phosphate isomerase is able to set a rapid balance between dihydroxyacetone phosphate and glyceraldehyde-3-phosphate formed by aldolase during glycolysis, which is associated with the pentose phosphate pathway and lipid metabolism via triosephosphates. Co-activation of triose phosphate isomerase and GAPDH accelerates glycerol transformation to pyruvate (Figure 4D). In this study, in the proteomic profile of the culture grown under alkaline stress, the mitochondrial porin VDAC (YALI0E12134p, mitochondrial porin (No. 20, Table 3) increased 2.2-fold (Figure 3). This is in good agreement with our previous studies performed using *Yarrowia lipolytica Polf* [12]. 

VDAC porins are known as conservative mitochondrial proteins regulating metabolite transport between the mitochondria and cytoplasm under both physiological and stress conditions [26,27]. VDAC is also involved in the regulation of respiration, ROS homeostasis, and yeast stress tolerance. The *Saccharomyces cerevisiae* mutants in the gene encoding mitochondrial porin synthesis are hyper sensitive to assimilating non-fermentable substrates (i.e., glycerol) due to the decreased respiratory activity [28]. VDAC is shown to play a positive role in carbon/energy stress signaling by providing nuclear location of AMPK/Snf1 signal transduction factors. The protein kinase Snf1 belongs to the family of AMP-activated protein kinases (AMPK). The positive regulation of AMPK/Snf1 with the VDAC participation was demonstrated using the *Saccharomyces cerevisiae* yeast. However, the conservatism of these proteins in eukaryotes suggests extending the phenomenon to some other organisms [29]. VDACs porins also provide the synthesis of cardiolipin and phosphatidylethanolamine in the inner mitochondria membrane, by transporting their precursors, namely phosphatidic acid and phosphatidylserine [30]. The increase in the porin level correlates with the rise in phoshatidic acid’s share in membrane lipids, which increases by 1.6 times at an alkaline pH [4]. The porin closely interacts with YALI0F31207p, YALI0B10362p, and YALI0A07084p proteins, which possess porin activity and import proteins into mitochondria. It also acts with mitochondrial GTPase and mitochondrial membrane ATP synthase involved in mitochondria transport and energy synthesis (Figure 4E). 

At an alkaline pH, in the yeast proteomic profile, the share of malate dehydrogenase YALI0D16753p (No 21) involved in carbohydrate metabolism (Table 3) also increased. Malate dehydrogenase, being a mitochondrial protein, catalyzes the conversion of malate to oxaloacetate, forming NADH in the tricarboxylic acid cycle [31]. It increased 2.9-fold when grown at an alkaline ambient pH (Figure 3). Malate dehydrogenase activation can accelerate the formation of citrate with a concurrent shift to lipid synthesis. The STRING database confirms this speculation by presenting the data regarding the close relationships of the enzyme with citrate synthase YALI0E02684p, citrate synthase, YALI0E00638p, and fumarate hydratase YALI0C06776p (Figure 4F). The malate dehydrogenase induction indicates the key role of mitochondria in metabolic readjustment and active VDAC participation in the adaptation to alkaline stress.

### 3.2. Changes in the Proteomic Profile of the Culture Grown under Heat Stress

The growth of the yeast under the increased (up to 38 °C) growth temperature led to the evanescence of both zone 5 (mannoprotein (Pir1, YALI0B20306p) and zone 23 (1,3-beta-glucanosyl-transferase YALI0B03564p), similarly to those in the alkaline conditions. Perhaps the proteomics changes are associated with the glycosidic bonds thermolability under those conditions. There was also revealed no VDAC porin (zone 20) in the profile. Thermal stress promoted the synthesis of 12 authentic proteins, and several proteins with chaperone features were substituted for some others (Figure 1 and Appendix A, Table 5). Thus, in these conditions, we could note a few new proteins, namely 42 (YALI0D22352p, SSA4 heat shock protein), 43 (CPAR2_700380, heat shock protein 70 family), 44 (YALI0E35046p, SSA4 heat shock protein), and 53 (YALI0D22352p, SSA4 heat shock protein) (Table 5). According to the STRING database, SSA4 heat shock protein of the HSP70 family (No. 42) could interact with the proteins with chaperone properties of YALI0F00880p and YALI0C07953p, which are responsible for the cell response to heat stress (Figure 5A). Moreover, within their interactions, there are the proteins of the YALI0C08987p and YALI0A00594p HSP families (Figure 5A), which are also in charge of cellular protein structural and functional integrity (Appendix A). In addition, among interacting proteins, clathrin was identified, which participates in clathrin-mediated endocytosis by absorbing the substances during the process and in the induction of the transmembrane carriers of amino acids, glucose, ions, etc. [32]. 

Growth at an elevated temperature induced the expression of some new proteins involved in ribosomal synthesis (48, Q0ZIC1_CANPA, GTPase cytoplasmic elongation factor 1 alpha) and the structural organization of the cell, in particular, cofilin interacting with actin (45, YALI0F20856p, Cofilin) (Appendix A). In addition, we obtained some antioxidant defense proteins (52, CPAR2_200490, Thioredoxin) and three formate dehydrogenases (46, CPAR2_203450; 50, YALI0B22506p (fragment), and 51, CPAR2_203450) (Appendix A). The cofilin family contains the most important actin-binding proteins, which, being the main regulators of actin dynamics (severing, depolymerizing, nucleating, and bundling activities), possess a high level of conservatism in all eukaryotes, beginning from lower ones to mammals [33,34]. An essential cytoskeleton readjustment is likely to happen under the conditions of heat stress. The activity of cofilin is known to be influenced by redox modifications of Cys residues via disulfide bonds, S-glutathionylation, and S-nitrosylation [34]. The oxidized cofilin is dephosphorylated at Ser-3, impairing its actin-depolymerizing function. Consequently, cofillins play multiple roles in the cells through various pathologies at high temperatures.

Fructose-biphosphate aldolase (zone 47, CPAR2_401230), participating in glycolysis and gluconeogenesis, and an uncharacterized protein of 49 (CPAR2_802980) were also expressed in these conditions. There was no ubiquitin-60S ribosomal protein L40 fusion protein, which supports the structural organization of ribosomes in the cells. Evidently, the expressed chaperones perform this function. Previously, we showed that the culture growth of 38 °C was accompanied by dramatic changes in the soluble carbohydrates composition. Trehalose substituted the dominating carbohydrate of mannitol [4]. In the paper by [35], the authors showed that the *Y. lipolytica* yeast needs trehalose to survive at an increased temperature. The changes in the carbohydrate composition correlate with the changes in the yeast proteomic profile. The induction of fructose-bisphosphate aldolase (No. 47, Table 5) in the yeast growing in glycerol containing medium indicates the activation of gluconeogenesis needed for the trehalose synthesis. It is noteworthy to remark that although trehalose inhibits some glycolytic enzymes’ activity in vitro, the enzymes’ association removes this effect under oxidative stress [36]. Some decrease in the antioxidant enzymes’ expression under heat stress is most likely to be partially counteracted by trehalose, possessing antioxidant activity. The expression of thioredoxin (No.52, Table 5) in the *Y. lipolytica* cells grown at an elevated temperature is apparently due to the cellular antioxidant response. The thioredoxin system includes three interacting proteins, namely thioredoxin, thioredoxin reductase, thioredoxin peroxidase, and NADPH as a proton source [37]. 

According to the STRING database, thioredoxin interacts with glutathione reductase, sulfiredoxin, and Cu-Zn SOD (Figure 5B), in addition to the thioredoxin system (thioredoxin reductase). This confirms the high adaptive response of the cells to oxidative stress under these conditions. The induction of NAD^+^-dependent formate dehydrogenases and some other D-2-hydroxyacid dehydrogenases is evidently related to an increase in catabolic processes and the activation of one-carbon residue metabolism (Appendix A). The formaldehyde expression in the cells appears due to the DNA demethylation, catabolism of serine, methionine, and purine bases [38]. Formate dehydrogenases are widely distributed in nature. Formaldehyde is one of C-1 metabolism’s products (one-carbon metabolism), and can form due to the catabolism of serine, methionine, and purine bases (1C units from serine to formate, and onto purines, thymidylate (dTMP), and methionine). DNA demethylation facilitates either formaldehyde or methanol, and formate dehydrogenase substrate expression. Functioning together with NAD^+^-dependent formaldehyde dehydrogenase and NAD^+^-dependent alcohol dehydrogenase, the enzymes contribute a lot to the NADH production via oxidizing the corresponding substrates, namely formate, formaldehyde, and methanol [39]. Therefore, we assume that the replenishment of the reducing equivalent pool could be used in the gluconeogenesis needed for the trehalose synthesis [4]. 

### 3.3. Changes in the Cell Proteomics during Culture Growth under Combined Stress

The assay of the DE gels showed the expression proteins typical to heat and combined stress (in the list, numbers 54 and 61 (CuZn SOD, CPAR2_500390, 2), 43 and 63 (30 kDa fragments of a protein of the HSP70 CPAR2_700380 family). Combined stress induced the synthesis of nine original proteins including two proteins involved in the translation process, namely the elongation factor (No. 57, CPAR2_20706) and protein 62 (SBDS domain-containing protein) needed for ribosome assembly (Table 6, Appendix A).

Among the proteins of energy metabolism, the ATP synthase subunit (No. 55, CPAR2_503440) and the pyruvate dehydrogenase subunit (No. 65, CPAR2_402950) were revealed (Appendix A). Moreover, a 12 kDa fragment of 36 kDa glyceraldehyde-3-phosphate dehydrogenase CPAR2_808670 was identified (No. 60), and two 12 kDa fragments of two different carbonyl reductases, CPCR1 and CPAR2_502580 (zones 56 and 58) were detected (Table 6). Nucleoside diphosphate kinase CPAR2_101390 (No. 59) and 22 kDa chaperone HSP70 Ssa2 HSP70 (zone 66) were two other original proteins (Table 6). It is noteworthy that under the conditions of alkaline pH and elevated temperature, the culture growth was accompanied by the changes in the proteomics, which were different from those at an alkaline pH and elevated temperatures. There was a change in the chaperone compositions (zones 63, 64, 66), there were no chaperone proteins either at an alkaline pH (No. 26, 27, 35, 41) (Table 2) or at an elevated temperature (37 °C) (No. 43, 44, 53) (Table 5). All the new chaperones identified belonged to the heat shock protein 70 family. 

Furthermore, several new spots were identified as carbonyl reductases (No. 56, 58), which belong to a large family of alcohol dehydrogenases and specialize in the asymmetric reduction of numerous compounds, namely aliphatic and aromatic ketones and diketones, keto acids, esters, and amides. Carbonyl reductases are of high stereoselectivity and are preferable as the NADH cofactor, being more thermostable than NADPH. Unlike most substrates, it is very stable in the presence of low-molecular-weight alcohols, such as isopropanol [40]. For example, carbonyl reductase isolated from *Candida parapsilosis* (CPCR; EC 1.1.1.1) could catalyze the asymmetric reduction of aliphatic and aromatic ketones and diketones [41]. The identification of several zones corresponding to the carbonyl reductases indicates the induction of catabolic processes related to scavenging the damaged proteins and lipids under combined stress (Appendix A).

The repair of nucleic acids and the maintenance of the apparatus for the new proteins’ synthesis play the crucial role in the yeast adaptation to the elevated temperature under alkaline conditions. Evidently, it leads to an increase in the SBDS domain-containing protein (No. 62) share, elongation factor 1 (No. 57), and nucleoside diphosphate kinase (No. 59) (Table 6, Appendix A). Upon growth at an alkaline pH and/or at an elevated temperature, the proteins of glycolysis/glyconeogenesis, which are involved in redistributing metabolic fluxes of carbon substrates, altered their composition (Appendix A). Under the conditions of combined stress, the ratio of the carbohydrate metabolism proteins also changed. The increase in the glyceraldehyde-3-phosphate dehydrogenase (No. 60) and pyruvate dehydrogenase (No. 65) levels (Table 6) is supposed to redirect pyruvate to fatty acid synthesis. Interestingly, previously, we showed that under combined stress, triglycerides were converted into di- and monoglycerides with the background of the storage lipids’ decrease compared to those under the normal growth conditions [4]. This may have occurred because the cells are in great demand for intensive protein synthesis and the energy for the synthesis under the conditions of oxidative stress (Appendix A). The increased expression of ATP synthase subunit 5 (No. 55) (Table 6) providing these processes partly confirms the hypothesis [42].

The comparison of the expression protein profiles with regard to their functional properties in the *Y. lipolytica* yeast grown under various conditions was performed. The protein spots included a mixture of several proteins and also contained short fragments of some known proteins. It is noteworthy that the CPAR2 proteins, according to the Matrix Science database, are classified as belonging to the *C. parapsilosis* yeast. Under stress conditions, the *Y lipolytica* culture is likely to synthesize some unknown forms of these proteins, which are closer to those from *C. parapsilosis*. Thus, we decided to compare the identified proteins according to their functions. Figure 6 shows the results of the comparison presented as the Venn diagram. Five proteins from the culture grown in the normal conditions proved quite original. Among them, there are two proteins of oxidative phosphorylation, namely *β*-subunit of ATP synthase (YALI0F03179p) and cytochrome c oxidase (cytochrome c oxidase); two structural proteins, i.e., the cell wall mannoprotein Pir1 (YALI0B20306p) and profilin (YALI0B07183p); and the thiol-specific peroxidase (YALI0A19426p) obligatory to maintain cell redox homeostasis. Such a set of proteins can be associated with active processes of cell growth and fission.

Under the conditions of alkaline pH, nine proteins were quite original. Two proteins involved in the ubiquitination process were identified, namely the SCF subunit of the ubiquitin ligase E3 complex (YALI0A10879p) and the endopeptidase activator (YALI0B09339); some proteins of the HSP family, i.e., small 20-kDa HSP34 and 70-kDa HSP chaperone SSA1 (YALI0C17347p), and the *β*-subunit of the immature polypeptide-associated complex (YALI0F08393p). All the proteins identified perform protein quality control [42], which is essential to maintain the prolonged survival of the culture under stress. The NADPH dehydrogenase (YALI0B07007p) involved in the purine metabolism, bis (5′-adenosyl) triphosphatase (YALI0E32736p), chain IV of cytochrome c oxidase (YALI0E19723p), and myosin light chain (YALI0E0E033) are the four remaining proteins detected under alkaline stress (Figure 4). Among common proteins for the normal and alkaline conditions, there were eleven proteins. The expression of four of them decreased. Two of these are the proteins of oxidative phosphorylation, namely the δ-subunit of ATP synthase (YALI0D22022p) and subunit VI of cytochrome c oxidase (YALI0E10144p). 1,3-beta-glucanosyltransferase (YALI0B03564p) and ubiquitin-60S, involved in carbohydrate metabolism, are the other two proteins. The expression of seven proteins increased under alkaline stress (Figure 4). Among them, there are VDAC (YALI0F17314p), two folding proteins, namely peptidyl-prolyl-cis-trans isomerase (YALI0C10230p), and 60 kDa-HSP chaperone (YALI0F02805p). Alkaline stress also caused the increased expression of 40S ribosomal protein S14 (A0A1H6Q0M6), tropomyosin (YALI0F27049p) requisite for the cell fission and transport, triose phosphate isomerase (YALI0F05214p) involved in glycolysis, and the TCA enzyme of malate dehydrogenase (YALI0D16753p) (Figure 4). Under alkaline stress, the changes in the proteomic profile are restricted due to the key cell demands, namely, on the one hand, to reduce the damage by ROS, performed either through the regulation of cytochrome-c- oxidase using subunits IV and VI [43] or through switching of the metabolism to glycolysis and lipid β-oxidation. VDAC plays the key role in this process [44]. On the other hand, the process of the protein quality control, during which de novo synthesized proteins are checked at different levels of structural organization, and those with any errors should be destructed, is of great importance. This process helps the yeast to resist the replicative aging under unfavorable conditions [45].

Under heat stress, four original proteins were found, namely formate dehydrogenase (as two different fragments YALI0B22506p and the complete protein of CPAR2_203450), fructose biphosphate aldolase (CPAR2_401230), disulfide oxidoreductase (CPAR2_106080), and cofilin (YALI085). Formate dehydrogenase oxidizes formate to CO_2_, simultaneously reducing NAD^+^ to NADH. The significance of the process in the protection of the cell against oxidative stress has been shown for some yeasts and plants, recently [45]. Disulfide oxidoreductase and an increment in glycolysis importance can also reduce the deleterious action of the ROS. HSPs of 70 kDa perform numerous functions in response to stress, protecting the proteins from any damaging impacts (folding, disaggregation, and degradation of injured proteins).

Under heat and alkaline stress, common 70 kDa HSPs from the SSA4 family were revealed. It is noteworthy that under alkaline stress, the YALI0D08184p protein was detected, while under heat stress, the YALI0D22352p and YALI0E35046p fragments were detected. Moreover, under both types of stress, there were some uncharacterized proteins, namely YALI0D00451p, YALI0F08327p, and YALI0A10747p under alkaline stress; and CPAR2_802980 under heat stress.

Under combined stress, we detected four original proteins. Firstly, there were CPCR1 (fragment) and CPAR2_502580 carbonyl reductases. Carbonyl reductases are known to be involved in the detoxification of cytotoxic products of lipid peroxidation [46,47]. ATP synthase subunit 5 CPAR2_503440, which contains SBDS-domain protein CPAR2_406210 involved in the ribosomes genesis, and a fragment of pyruvate dehydrogenase CPAR2_402950, were other original proteins for the combined stress.

The *α*-1 elongation factor of the cytosolic GTPase was the only common protein for heat and combined stress. Moreover, under heat stress, it was the Q0ZIC1_CANPA protein, while under combined stress, it was the CPAR2_207060 protein fragment. We detected no common proteins for the alkaline and combined stresses. Of note, SSA4, revealed under both thermal and alkaline stress, was not detected under the conditions of combined stress. Two proteins, namely glyceraldehyde-3-phosphate dehydrogenase (YALI0C06369p in the normal and alkaline conditions and CPAR2_808670 under combined stress), as well as nucleoside-diphosphate kinase (YALI0F09229p in the normal and alkaline pH conditions, and fragment CPAR2-101390) were common for the normal, alkaline and combined stress conditions. HSPs of 70 kDa, namely CPAR2_700380, detected under combined stress, and the YALI0F25289p fragment, detected under normal conditions, were the common proteins for the normal and the combined stress conditions. Thioredoxin was the common protein revealed under normal conditions and both types of stress. CPAR2_200490 was detected under heat stress and YALI0F01496p was detected at normal and alkaline pH values. Moreover, the level of the latter protein decreased under alkaline stress.

Finally, Cu/Zn-SOD was detected under all the conditions studied. Under normal conditions and at an alkaline pH, YALI0E12133p was the common protein, and its level increased at pH 9.0. Under heat stress, we detected CPAR2_500390, while under combined stress, it was identified only as a fragment. Thus, Cu/Zn-SOD, thioredoxin, and various 70 kDa HSP chaperones proved to be the most versatile cytoprotective proteins. The regulation of cytochrome c oxidase and an increase in the VDAC amount were the adaptation factors original for the alkaline stress. Upon adapting to the heat shock, formate dehydrogenase was an exclusive protein, while under the combined stress conditions, it was carbonyl reductase.

## 4. Concluding Remarks 

The results show that the proteomic profile reflects the metabolic readjustment under various growth conditions (Figure 1). At an extreme pH, pathways shift towards the oxidative metabolism of glyceraldehyde-3-phosphate into pyruvate in the mitochondria, its implication in the Krebs cycle followed by forming citrate, its transport into the cytosol, and inclusion into lipid biosynthesis (Figure 1). At an increased temperature, glycerol is metabolized, forming glucose in the cytoplasm with a concurrent synthesis of trehalose (Figure 1).

Therefore, the analysis of the protein composition of the *Y. lipolytica* yeast under different conditions revealed some regularity in the response to various stresses. We observed a special impact of alkaline pH and increased temperature on changing the proteomic profile. The changes in the mitochondria activity, which were displayed in inducing VDAC porin and malate dehydrogenase, played an essential role in the adaptation. The culture growth under combined stress caused the changes in the proteomics differed from those at either alkaline or heat stresses alone. Moreover, the readjustment of the protein (chaperones and carbonyl reductases) composition and amount occurred, which indicated the induction of catabolic processes related to scavenging damaged proteins and lipids under the conditions of combined stress. The data analysis suggests that adaptation to the combined stress should include significant readjustment of the carbohydrates and lipid metabolism, as well as the cell’s protection against lipid peroxidation products and protein destruction. The combination of chronic alkaline and heat stress (pH 9.0, 38 °C) leads to the cross-adaptation expressed in “switching” usual metabolism over the adaptation to the most damaging stress factor, i.e., the increased temperature.

Thus, modeling the cultivation conditions of the *Y. lipolytica* yeast may be used for the streamlined synthesis of the lipid or carbohydrate metabolites for industrial purposes.

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
