# Peer review of "Proteomics Readjustment of the Yarrowia lipolytica Yeast in Response to Increased Temperature and Alkaline Stress"

_microorganisms, 2021, doi:10.3390/microorganisms9122619_

Round 1

Reviewer 1 Report

The manuscript by Sekova et al. gives a very interesting insight in the proteomics of Y. lipolytica during stress conditions. The aim of the study itself is generally sound, but the manuscript needs improvement before acceptation. The manuscript has three weak spots, which should be reinforced: language, pictures, and Concluding remarks.

Recommendations:

  1. Major language correction is needed.
  2. Quality (resolution) of pictures must be increased.
  3. The section Concluding remarks includes pictures which are nice but not at the right place. If the authors feel they have to put picture in this section, maybe it could be better to show the affected metabolic pathways on a metabolic map and leave Fig. 7 and Fig. 8 to Supplementary materials.

Author Response

Dear Reviewer!

We try to answer all of your comments and remarks.

The Authors

The manuscript by Sekova et al. gives a very interesting insight in the proteomics of Y. lipolytica during stress conditions. The aim of the study itself is generally sound, but the manuscript needs improvement before acceptation. The manuscript has three weak spots, which should be reinforced: language, pictures, and Concluding remarks.

Recommendations:

We are grateful to the Reviewer for the remarks and recommendations on improving the manuscript

  1. Major language correction is needed.

We have checked and the English language and corrected the mistakes

  1. Quality (resolution) of pictures must be increased.

We have improved the pictures to the maximal resolution the Program permits.

  1. The section Concluding remarks includes pictures which are nice but not at the right place. If the authors feel they have to put picture in this section, maybe it could be better to show the affected metabolic pathways on a metabolic map and leave Fig. 7 and Fig. 8 to Supplementary materials.

We have removed Figures 7, 8 and 9 in the Supplementary section, now they are Fig. S2, S3 and S4, respectively. We added the Scheme with metabolic pathways under different conditions.

Reviewer 2 Report

The paper overall is well-written, and results are clearly shown. However, the network analyses in figures 4 and 5 are pixelated/fuzzy in my copy. This must be fixed in final version. The one area that I think could be improved on is what the purpose/relevance of this knowledge is. For example, what can be done with it? Is there any industrial or medical usefulness for these results. 

Author Response

Dear Reviewer!

We are grateful to you for valuable remarks and comments.

Thank you a lot, the authors

The paper overall is well-written, and results are clearly shown. However, the network analyses in figures 4 and 5 are pixelated/fuzzy in my copy. This must be fixed in final version. The one area that I think could be improved on is what the purpose/relevance of this knowledge is. For example, what can be done with it? Is there any industrial or medical usefulness for these results. 

Answers to reviewer

We are grateful to the reviewer for the remarks and recommendations on improving the manuscript

We have improved the quality of the images in Fig. 4 and 5 to the maximal resolution the Program permits. In the concluding section, we added the metabolic scheme of glycerol metabolism readjustment in the Y. lipolytica at heat and pH stress with a possible use in the industry.

Reviewer 3 Report

The work presented by the authors aims to investigate the different proteomic profile expressed by the yeast Yarrowia lipolytica in different growth conditions. The work presents some aspects to be improved before being taken into consideration for publication.

Some of my suggestions:

1) the introduction is exceedingly long and dispersive. It should be better centered on the purpose of the job.

2) the results and discussion paragraph needs improvement. It is a bit confusing and there are several errors:

- non-correspondence of the number of proteins between the text and the tables. For example, Table 1: in the text they spoke of five proteins but in the table, there are six. Similar errors also in other tables such as 2.1 and 2.2.
- in the legend of Fig. 3 the authors reported "Each value is the mean ± SD of three analytical assays" but in the graph the SD are not reported. Furthermore, the authors spoke of "spot intensity comparison" but there is no statistical analysis. This figure needs to be modified and appropriate statistical analysis added.
- I would suggest dedicating a paragraph to the discussion, in this format it is not remarkably effective.
3) The conclusion is too distracting. It still contains a discussion part and other images. Authors should rewrite the conclusion emphasizing the strengths of their work and the implications it entails.
4) figures. the figures are many. It would be helpful for the manuscript to move some figures in the Supplementary Information also according to the new draft of the work. I would eliminate or move the figures from the conclusion. If the authors found them useful, Figures 7-9 should be moved to results. The conclusion should be concise and useful to focus the data obtained. 

Minor errors:
in abstract: add the full name of the yeast Yarrowia lipolytica

in materials and methods: "Yeast Strains" should be changed to "Yeast Strain" because only one strain (Wild-type Yarrowia lipolytica W 29) is under consideration.

Author Response

Dear Reviewer!

Thank you very much for your comments and recommendations.

Sincerely yours, the authors

The work presented by the authors aims to investigate the different proteomic profile expressed by the yeast Yarrowia lipolytica in different growth conditions. The work presents some aspects to be improved before being taken into consideration for publication.

Some of my suggestions:

Answers to reviewer

We are grateful to the reviewer for the remarks and recommendations on improving the manuscript.

  • the introduction is exceedingly long and dispersive. It should be better centered on the purpose of the job.

Answer: We have shortened the Introduction and focused it on the purpose of the study 

2) the results and discussion paragraph needs improvement. It is a bit confusing and there are several errors:

- non-correspondence of the number of proteins between the text and the tables. For example, Table 1: in the text they spoke of five proteins but in the table, there are six. Similar errors also in other tables such as 2.1 and 2.2.

Answer: We have improved the section. We have corrected the data of the text and the Tables

- in the legend of Fig. 3 the authors reported "Each value is the mean ± SD of three analytical assays" but in the graph the SD are not reported. Furthermore, the authors spoke of "spot intensity comparison" but there is no statistical analysis. This figure needs to be modified and appropriate statistical analysis added.

Answer: The corrections in Fig.3 were made

- I would suggest dedicating a paragraph to the discussion, in this format it is not remarkably effective.

Answer: The data in Fig 3 were analyzed and described on pages 10, 14 (it is highlighted with blue). 

The conclusion is too distracting. It still contains a discussion part and other images. Authors should rewrite the conclusion emphasizing the strengths of their work and the implications it entails.

4) figures. the figures are many. It would be helpful for the manuscript to move some figures in the Supplementary Information also according to the new draft of the work. I would eliminate or move the figures from the conclusion. If the authors found them useful, Figures 7-9 should be moved to results. The conclusion should be concise and useful to focus the data obtained. 

Answer: We have transferred Figures 7, 8 and 9 in the Supplementary section. In the Concluding section, we added the metabolic scheme of glycerol metabolism readjustment in the Y. lipolytica at heat and pH stress with a possible use in the industry.

Minor errors:
in abstract: add the full name of the yeast Yarrowia lipolytica

Answer: Done

in materials and methods: "Yeast Strains" should be changed to "Yeast Strain" because only one strain (Wild-type Yarrowia lipolytica W 29) is under consideration.

Answer: Done

Round 2

Reviewer 3 Report

I thank the authors for the changes made. The manuscript seems to me to have improved. I have just one last tip. In Figure 3, since the authors described "comparison of spot intensities" and different quantities in terms of abundance, I would suggest that the authors add the statistical analysis (if not performed) between the two pH conditions and the corresponding p-value.

Author Response

We thank the Reviewer for valuable remarks. 

We have corrected Fig 3 according to your comments.